# A Cost–Benefit Analysis Simulation for the Digitalisation of Cold Supply Chains

**DOI:** 10.3390/s23084147

**Published:** 2023-04-20

**Authors:** Oliver Schiffmann, Ben Hicks, Aydin Nassehi, James Gopsill, Maria Valero

**Affiliations:** 1Department of Mechanical Engineering, University of Bristol, Bristol BS8 1QU, UK; 2Centre for Modelling and Simulation, Bristol BS16 7FR, UK

**Keywords:** prediction, optimisation, simulation, digitalisation, Industry 4.0

## Abstract

This paper investigates using simulation to predict the benefits and costs of digitalising cold distribution chains. The study focuses on the distribution of refrigerated beef in the UK, where digitalisation was implemented to re-route cargo carriers. By comparing simulations of both digitalised and non-digitalised supply chains, the study found that digitalisation can reduce beef waste and decrease the number of miles driven per successful delivery, leading to potential cost savings. Note that this work is not attempting to prove that digitalisation is appropriate for the chosen scenario, only to justify a simulation approach as a decision making tool. The proposed modelling approach provides decision-makers with more accurate predictions of the cost–benefit of increased sensorisation in supply chains. By accounting for stochastic and variable parameters, such as weather and demand fluctuations, simulation can be used to identify potential challenges and estimate the economic benefits of digitalisation. Moreover, qualitative assessments of the impact on customer satisfaction and product quality can help decision-makers consider the broader impacts of digitalisation. Overall, the study suggests that simulation can play a crucial role in facilitating informed decisions about the implementation of digital technologies in the food supply chain. By providing a better understanding of the potential costs and benefits of digitalisation, simulation can help organisations make more strategic and effective decisions.

## 1. Introduction

Supply chains often reach a complexity where their operation cannot be predicted with absolute accuracy. The distribution of perishable goods is one such example, where stochastic events such as unexpected spoilage, unavoidable delays, and the over-consumption of resources (fuel, tyres) cause unintended wastage [1]. An expanding population, increasing demand for product diversity, and critical climate issues are forcing the industry to search for methods that can reduce wastage.

Perishable goods such as food, pharmaceuticals, and biological material are often transported using cold supply chains (CSCs). Wastage arises as a function of travel time, the temperature profile experienced by the perishable good, and a degree of randomness that cannot be accounted for by the travel time and temperature profile. In some cases, there exists legislation or regulations that need to be adhered to if one wishes to sell the goods on-wards (e.g., pharmaceuticals) [2].

Industry 4.0 would advocate the Internet-of-Things (IoT) digitalisation of the supply chain, with vehicles fitted with global positioning (GPS) and environmental (e.g, temperature and humidity) sensors that continuously transmit data to the cloud where it can be analysed, and actions taken. For instance, if a temperature excursion occurs during transportation, the system could alert relevant parties, enabling them to take corrective action before the goods are compromised [3]. This early detection is key to minimizing the spoilage, degradation, or contamination of the product, which could result in significant economic losses or harm to human health as well as allow meeting regulatory requirements. Digitalisation of CSCs is also a key step towards Logistics 4.0, and much of the work in this paper is relevant to the opportunities and challenges presented in this field—food logistics, resource planning, and transportation management [4].

However, the digitalisation of a CSC comes at a potentially prohibitive cost, both capital, such as sensor and network connectivity hardware, and operational, such as network bandwidth, storage, analytical processing, software licensing and personnel, and expenditure. This is particularly challenging for small to medium-sized enterprises (SMEs). Nonetheless, digitalisation may not be appropriate for all stakeholders: For instance, in the UK alone, small to medium-sized enterprises (SMEs) account for 97% of organizations involved in the UK food supply chain [5]. While digitalisation can provide benefits such as increased efficiency and reduced waste, the costs associated with implementing digitalisation may be too high for SMEs. Additionally, many SMEs may lack the expertise or resources to effectively implement and manage digitalisation systems.

The problem of weighing the costs and benefits of digitalising an existing service is complicated further by the fact that not all factors are easily calculated prior to adoption (Figure 1). This research investigates how simulation can be used to evidence the stochastic/variable costs and benefits when digitalising a CSC. The resulting simulation could subsequently be used to help make recommendations to the degree of digitalisation under current economic conditions. Clearly, digitalisation has strong potential for improving CSCs, but it is unclear whether SMEs in particular stand to benefit. Hence, this work develops an approach for determining the outcome of digitalisation before implementation. Knowledge of the outcome is useful to SMEs, beneficial or not, as it can help to avoid poor investments.

The paper continues by exploring related work to determine a method for making digitalisation recommendations (Section 2). This is followed by the simulation approach taken to evidence the cost–benefits of digitalising CSCs (Section 3). The simulation was then applied to a case study of cold beef distribution within the UK (Section 4 and Section 5). The results of these simulations are then presented (Section 6). A discussion then ensues as to the role of simulation in supporting digitalisation decision-making, accompanied by example payback time calculations using simulation results (Section 7). The paper then concludes with the key findings from the study (Section 8).

## 2. Background

Cost–benefit analysis (CBA) is a widely used method for evaluating the financial feasibility of projects and informing decision-making processes. CBA quantifies and compares expected costs and benefits of a project over a specified time period to determine whether the benefits outweigh the costs and therefore whether the project is worthy of investment [6]. In this way, CBA can help evaluate and understand the potential impacts and opportunities for digitalising business operations [7]. With more accurate assessments of the expected benefits and costs, organizations can make more informed decisions about digitalisation [7] and thus implement more effective strategies

In CBA, benefits and costs can be classified as either estimable or stochastic. Estimable costs and benefits refer to those that can be accurately predicted or estimated based on relevant information and data [6], whereas stochastic ones refer to those that are uncertain and subject to random variation [8]. Stochastic costs and benefits help to capture the randomness of project outcomes, providing a more realistic representation of the potential financial risks and opportunities associated with a project [6].

The incorporation of stochastic costs and benefits in CBA provides a more robust and comprehensive assessment of a project’s financial potential and/or performance [9]. Simulating and estimating variable or stochastic costs and benefits in CBA has become increasingly popular in recent years. Rather than being offered a single value assessment of CBA, decision-makers are now being provided with the landscape of opportunity, enabling them to consider the likelihood of certain outcomes and the potential consequences of different scenarios [8]. Evaluating the landscape is particularly important for complex systems where volatility in their operation exists, such as CSCs.

The most cost-effective way to transport goods between locations should be determined by taking an array of estimable and stochastic factors into account. Figure 1 shows the array of intangible cost and benefit factors when considering the digitalisation of CSCs. These include increased customer satisfaction (e.g., reduced waiting times) and thus brand recognition to opportunities lost (e.g., sub-optimal processes and efficiencies).

Therefore, the capability to simulate and thus estimate stochastic benefits and costs as accurately and/or realistically as possible cannot be underestimated. The role of simulation to provide insights into the cost and benefits has seen success in evaluating the digitalisation of sectors, such as healthcare and infrastructure, but has yet to be explored in the digitalisation of CSCs [7,10,11].

### 2.1. Modelling Supply Chains

Simulation is an ideal tool for analysing systems where in situ testing of proposed adjustments is unfeasible [12]. Numerous types of models have been developed to support supply chain design and analysis (spreadsheet simulation, system dynamics, discrete-event simulation, and business games) [13]. This poses the question, what type of model is most appropriate for examining the CBA of digitalisation in CSCs? Existing work has identified the CSC as unique, due to the intrinsic link between product quality and supply chain design [14]. This work presents discrete event simulation (DES) as an appropriate modelling technique for analysing the sequence of events within a supply chain. This is due to the complexity of the supply chain network preventing analytical evaluation, the ability of DES to incorporate uncertainty [15], and the frequent occurrence of individual events—such as the arrival of an individual order [13].

To consider interactions within an event, agent-based modelling is typically used [16]. This work uses an agent-based modelling (ABM) approach. ABM better enables the user to model interactions between the “agents” of a system. These agents can take almost any form. In this work, they are used to model the behaviour of individual trucks responsible for feedstock delivery. This work also boasts the benefit of combining an ABM approach with geographic information system (GIS). GIS offer a significant benefit in the context of this report, as they enable the accurate calculation of travel times.

### 2.2. Optimisation Methods for Supply Chains

Digital simulations are frequently coupled with computational optimisation techniques to test the potential benefits of suggested modifications [17]. Distribution strategy optimisation is a common problem encountered in various fields, and the increasing complexity and size of supply chains have made prior optimisation methods less favourable [18]. Recently, meta-heuristics have gained success in supply chain optimisation [19].

Optimal inventory and supply strategies are required for the supply of perishable goods, which necessitates balancing cost, efficiency, and responsiveness [20]. Several studies have explored the optimisation of supply chain operations during the distribution stage [21]. In a perishable goods supply chain, [21] applied a multi-objective optimisation model to distribution planning of a multi-echelon supply chain with deterministic demand, aiming to balance cost, delivery lead time, and cold storage setup cost. The optimal storage locations and shipping quantities were determined for two case study products, guava and lemons.

While much of the literature has focused on pre-planning of supply strategies [20], some studies have investigated how distribution can be modified during supply [22]. Ref. [22] aimed to enhance the delivery quality of perishable goods by increasing their safety through the use of radio frequency identification (RFID) technology. Real-time monitoring of product quality was made possible by RFID technology, which enabled adjustments to the distribution strategy at sequential stages of the supply chain. Consequently, a distribution-planning model was developed that could consider the logistics dynamics.

### 2.3. Proposed Approach

The reviewed literature suggests computer modelling as a suitable approach [12,23]. A numerical approach using computer simulation is a valuable tool for modelling complex systems, such as supply chains. In particular, when considering the possibility of re-routing carriers during delivery, the use of simulation offers several advantages over other modelling approaches. Firstly, simulation allows for the digital replication of the supply chain, providing a dynamic model that can be used to test and optimise different strategies. Secondly, simulation can provide a cost-effective means of evaluating different scenarios, allowing decision-makers to identify the most efficient and effective options. Furthermore, a simulation approach can enable the testing of different parameters, such as delivery routes, vehicle types, and scheduling options, to find the optimal solution. Ultimately, the use of a numerical approach using computer simulation offers a powerful and flexible tool for modelling supply chains and provides a means of evaluating and optimising the performance of these systems [24].

Previous literature has also identified dynamically re-routing trucks mid-delivery as a gap in the explored approaches for optimising supply chains [20,22]. Sensorisation could provide the data required to make decisions about dynamic re-routing. This would include data describing the environmental factors (temperature, humidity, oxygen/CO${}_{2}$ levels) of cargo, which could be fed into a predictive shelf life model. This would return a remaining shelf life for the cargo that, when compared with the remaining delivery times, could be used to make informed decision about the probability of cargo spoiling before its destination.

Note that there are other options for supply chain actuation that can be implemented to reduce wastage. These include:Dynamic pricing—By adjusting prices in real-time based on supply and demand, companies can better match supply with demand, reducing waste [25].Inventory management—By optimising inventory levels and reducing excess inventory, companies can reduce waste [26].Packaging design—By designing more efficient, sustainable, or smart packaging, companies can reduce waste in the supply chain [27].

However, this work investigates re-routing; as the logic required is contained wholly in the distribution stage of the CSC (reducing model complexity), it is an approach that can be implemented with relative ease (delivery drivers only need a new GPS destination), and it remains unexplored by CBA [20,22].

## 3. Modelling a General Digitalised Cold Distribution Chain

The review of existing work aided the development of an approach to evaluate the cost–benefit of digitalising CSCs, and this section details the requirements for modelling a digitalised cold distribution chain in non-software specific terminology. This includes the type of communication required between agents, how data would be used to optimise the distribution strategy, and the required behaviour for individual agents. The implementation is then described in Section 4.

### 3.1. Conceptual Model of a General Digitalised Cold Distribution Chain

This work explores increasing sensorisation in the distribution stage of a CSC and how collected data can be used to provide re-routing instructions to carriers. Figure 2 shows the conceptual model of a parametric digitalised CSC. Today’s commercially available connectivity capabilities enable data and knowledge to flow among Suppliers (e.g., producers, distribution centres, etc.), Carriers (e.g., trucks for land transportation, cargo planes for air transportation, cargo ships for sea transportation, etc.), and Customers (e.g., retailers, end consumers, etc.). Further, commercial sensing and actuation technologies enable two-way control and feedback of physical and digital spaces assets, processes, and models [28].

Information regarding key assets and processes in the *Physical space* is collected via *Edge Devices*. *Edge Devices* typically include an array of sensors measuring relevant physical and chemical parameters, such as temperature, light levels, humidity, etc., during transportation (*Carriers*) and/or warehousing (*Suppliers*). A microcontroller/hub collects and digitalises the sensed data, which are then sent to the *Internet/Cloud* using *Gateway* technologies and standards such as MQ Telemetry Transport (MQTT) standards or HTML or Extensible Messaging and Presence Protocols (XMPP). Communication technologies (Bluetooth, Wi-Fi, GPRS and NFC) can provide *Connectivity* capabilities between diverse devices.

The data are then stored (data storage) in relation, document, and/or graph databases such as MongoDB, MySQL, Cassandra and Neo4J. The data can then be retrieved for data mining/science activities. With numerous and diverse data from Suppliers, Carriers, and Customers (e.g., concerning to inventory, transportation and/or warehouse management), datasets are often large with complex information schemas resulting in a considerable computational overhead and knowledge burden.

Cloud computing can provide fast, ubiquitous, on-demand access to configurable computing resources [29]: from infrastructure and hardware assets (Infrastructure as a Service, IaaS) such as computer networks, servers, storage, etc., to software development environments (Platform as a Service, PaaS) for development of bespoke (data mining) applications. Using AI, machine learning-powered analytics (e.g., Apache Spark, Microsoft HDInsight, etc.) and measured Physical space data are used to build the corresponding models in the Digital space. These virtual models can be used to predict the current state and forecast the future state of the (physical) assets and processes. Such insights and foresights are used to inform decisions and actions to be performed in the Physical space (actuation data and knowledge streams), such as re-routing of Carrier vehicle(s), returning to base, etc. Software as a Service (SaaS) “ready-to-use” software applications can facilitate the management, control, and integration among key players in the cold distribution chain, so actuation can be performed or optimised based on diverse criteria, such as delivery time, driver and/or vehicle carrier availability, etc.

### 3.2. Model Operation

To model the system, the communication between agents enabled by digitalisation, the movement of carriers during delivery, and the logic governing actuation will all need to be captured. Stochastic events, such as spoilage, also need to be simulated.

The simulation approach taken was a multi-method approach combining discrete event simulation (DES) and agent-based modelling (ABM). DES handled the stochastic events within a simulation [14,15], such as spoilage during distribution, while ABM handled the communication and interaction between agents [16]. The agent types considered in this supply chain system, along with the communication between agents, is shown in Figure 3. Agents in the simulation are shown in blue, and dashed lines indicate communication that is enabled through digitization of the supply chain. The first line of information associated with each arrow represents the information passed during that communication. The second represents the trigger causing that communication. The green computer represents the optimiser responsible for adjusting the distribution strategy.

The simulation consisted of four agent populations:The Customer(s)The Supplier(s)The Carrier(s)The Order(s)

It should be noted that Figure 3 shows the information exchange between agents required for distribution strategy adjustment (GPS coordinates and shelf life of carried products). In the simulation approach, exchanging types of information depending on the digitalisation of interest is possible. However, the communication channels between agents would remain the same, and the modification to modelling approach would be minor.

Accompanying the communication pathways and agents is an Optimiser. This is not an agent within the model but rather an external process that one places functions that utilise the data captured from digitalisation and can offer insights and/or direct intervention into the CSC. For example, the Optimiser could analyse the data to minimise the required distance for delivery.

Carriers will pass data to a computational hub whenever their sensors take a reading (ignoring data loss, poor signal, and late data collection). This would then use an analytical model to convert the sensor readings (temperature, gas levels, etc.) into an expected shelf life for the perishable good being distributed. If this model predicts spoilage before the Carrier would arrive at the Customer, then the Optimiser would be activated. Since this event would occur unexpectedly (shipments that are expected to spoil would not leave the Supplier), it could be modelled using a stochastic trigger. Therefore the shelf-life model need not be considered in this modelling approach.

When this trigger occurs, the optimiser should be run. This would first check if there are any available options for re-routing, as this would be desirable to replacing the shipment. This would be achieved by iterating through the Customers with currently undelivered orders. If none exist, the only option is to return early and send another Carrier from the Supplier. If only one is within range, then the Carrier’s destinations are swapped (providing the non-spoiling Carrier can reach its new destination). If there is more than one option for swapping Customers, an algorithm optimising for minimum total required driving distance to achieve delivery should be run. The output of this algorithm would be a new distribution strategy represented by the destinations/Customers for each Carrier.

### 3.3. Agent Behaviour

The inter-agent communication shown in Figure 3 describes the higher level operation of the simulation model. The required logic for the Optimiser has also been described. Each agent must contain its own logic to govern its behaviour. This section details the required behaviour of each agent. Note that the Order agent has no logic in the approach described and would simply contain information about the Customer that created it. They enable traceability through the simulation and are of a fixed size.

#### 3.3.1. The Customer Agents

Figure 4 describes the customer agent statechart. Each Customer agents starts in the operating state simulating sufficient stock. A stochastic trigger would then indicate that stock has become low and prompt the customer to place an order, thereby creating an Order agent. The Customer agent would then await order confirmation. They would exist in this state when there are no Carrier agents available to complete their order. Once a Customer is in this state, the agent becomes available for swapping if a Carrier’s shipment is expected to spoil. Once a Carrier becomes available, they are awaiting delivery. They remain open for swapping in this state as well. Should their assigned Carrier agent spoil without option for re-route, their order would be cancelled, the Order agent removed, and they would place a new order. If their Carrier agent successfully arrives, they return to operating with sufficient stock and are no longer available for swapping.

#### 3.3.2. The Supplier Agents

The Supplier agents govern the Order agents created by the Customer agents and manage the fleet of Carrier agents. Their statechart is shown in Figure 5. The Supplier agent exists in a passive state until an Order agent is created. At this point, it begins iterating through the population of Carrier agents to check if any are available (parked and waiting at the Supplier agent’s location). If no Carrier agents are available, the Supplier agent waits for one to return. If multiple Carrier agents are available, one is selected at random. An available Carrier agent would be sent to the Customer agent specified by the Order agent. The Supplier agent then continues assigning orders as they arrive, listening for delivery confirmations. Note these confirmations are from the Customer and not the Carrier to enable swapping of Customers. Once the Supplier receives confirmation of a successful delivery, or of order cancellation, the Order agent is removed.

#### 3.3.3. The Carrier Agents

The statechart describing the behaviour of the Carrier agents is shown in Figure 6. A population of Carrier agents are created at the Supplier agent. Each Carrier agent would wait at the Supplier agent until an Order agent is created and given to them. Once a Carrier agent receives an Order agent, the Carrier agent starts to travel to the Customer agents’ destination. The Carrier agent can only be destined for one Customer at a time before returning to the Supplier. However, they are capable of changing Customers mid-delivery if the optimiser issues them a new Customer index. While in transit, a stochastic spoilage event may trigger causing the Carrier agent to enter the expected spoilage state and pass data about itself to the Optimiser. The Carrier agent will then momentarily wait to see if there are any re-route options available to it. If not, the Carrier agent will cancel the order and begin its journey back to the Supplier agent. If a re-route option does exist, then the Carrier agent will move to its new Customer agent. After arriving at a Customer agent, the Carrier agent will return to the Supplier agent.

Expected delivery time was not a parameter included in the model. Instead, a Carrier speed was selected and a route planned using the GIS methods. During model construction, delivery was seen to be several hours. As such, with an allowable delivery time of 6 days (explained in Section 5), beef was unlikely to spoil due to extended distribution. Therefore, the spoilage trigger is modelling unexpected product quality abuses. These are primarily temperature fluctuations caused by poor circulation of air, radiant heat from the floor, substandard insulated areas, and/or refrigerant leakages [30].

## 4. Using the Model to Evaluate Digitalisation of the UK Beef CSC

To demonstrate the model’s potential for evaluating CBA of digitalisation in a CSC, the model was used to evaluate the CBA of digitalisation of the UK Beef CSC. The model was instantiated using AnyLogic and Microsoft Excel.

AnyLogic provides an environment capable of supporting the multi-method approach. ABM would be required to accurately simulate the interactions between agents as well as govern their behaviour. DES would also be required to model the stochastic spoilage of cargo. AnyLogic also provides geographic information system (GIS) functionality, which is necessary to create the variety of re-route options needed to test mid-transit strategy adjustment. Excel provided the environment for the optimiser. It features both the capability and is a tool typically used by SMEs throughout the food supply chain. The benefits of these choices are summarised in Table 1 and Table 2.

### 4.1. Case Study—UK Beef CSC

Meat products are safety-critical foods (meaning deterioration in product quality renders the product unsafe) that have significant economic impact due to their high retail price, large ecological footprint (e.g., water and land usage), fuel consumption in transportation, and CO${}_{2}$ [28]. Additionally, the costs associated with meat are further compounded by its susceptibility to spoilage, which can have an adverse effect on both quality and safety. With the anticipated 12% rise in global meat consumption by the end of the decade [28], the distribution of meat appears an ideal candidate for digitalisation.

As a significant meat producer with a damp climate and rolling grasslands, the UK’s domestic market accounts for approximately 75% of the total meat production. For the purposes of this exploratory and simplified case study, Wales was chosen as the largest meat supplier to major UK cities [33] such as London, Liverpool, and Manchester, which are significant consumption centres. The AnyLogic simulation parameters, Table 3 and Table 4, were chosen based on the most recent available data from the UK, taking into account a trade-off between the model’s plausibility and simplicity to ensure simulation convergence. This case study is summarised in Table 3.

To model the supply chain specified in this section, the assumptions listed in Table 4 were made. The effects of these simplifications on the model outputs are discussed in Section 7.

While the class of problem considered in this paper is not ubiquitous, a number of cases that embody similar characteristics exist. These include distribution of other food stuffs, medicines (vaccines in particular), and biological material (blood, donor organs). All of these are highly distributed, time-quality dependant, and safety-critical. Therefore, this case study should be sufficient for identifying the advantages and limitations of the proposed approach.

### 4.2. Optimiser Implementation

The optimiser implemented in the case study aims to minimise the driving distance before delivery (but after leaving the Supplier) and when a spoilage event occurs, identifying whether re-route of an existing carrier could improve upon the minimum distance before delivery. The statechart for the Optimiser is shown in Figure 7. There is no trade off to consider when planning routing before leaving the Supplier. All Carriers are located at the Supplier so one Carrier uses GIS methods to choose the shortest path to the Customer and begins its journey.

On a spoil event, the Optimiser is triggered and performs the following steps:Computes the pair-wise distances between the carriers out on delivery and customers, and one additional carrier the will be leaving the distribution site.Determine the minimum distance travelled for carriers to reach customers.Re-assign carriers to customers (if required).

The evolutionary solving method (ESM) was applied to determine the optimal assignment of carriers to customers and eliminated the need to compute all permutations. To further improve the performance, the following steps were taken:Removing the spoiled Carrier being returned to the Supplier as a viable solution shrinks the solution space, making it easier/faster to find a globally optimal solution. This was done by checking if any other Customers with outstanding orders were even within range of the spoiling Carrier before running the optimiser.Applying integer constraints to decision variables and setting bounds based on the relevant Customers drastically reduces the solution space, improving the chance of finding a globally optimal solution. These bounds were set equal to the smallest and largest values found in the Customer’s with outstanding orders indices. For example, if there are 10 customers but only customers 2, 4, and 7 have outstanding orders, then the ESM sets the bounds as 2 and 7.Making population size a function of the number of decision variables improves the ESM by reducing the computational power spent on smaller scale problems, and improving the ability to find the global optimum for larger scale problems.

### 4.3. Performance Metrics

In selecting metrics to compare scenarios, the aim is to provide a holistic description of the supply chain such that the stochastic and intangible costs and benefits of sensorisation can be better predicted. The combination of ABM and AnyLogic used in this work makes collection of data describing the agents and distribution simple. Almost any data likely to be useful could be collected depending on stakeholder priorities. For this work, the key identified data include the percentage of product that is wasted during simulation, the distance covered per successful delivery, the amount of time customers spend waiting for an order, and the remaining shelf life of the product when it is delivered.

In the literature, an approach for measuring supply chain performance is described where performance measures are categorised by the resources, outputs, and flexibility of the supply chain [36]. The proposed approach uses miles driven and surplus shelf life as representative of distribution costs and delivered product quality, respectively. Flexibility is more difficult to measure, but monitoring the hours spent waiting should provide an indication of flexibility within the CSC. If the system can handle unexpected events that cause extended delivery times (spoilage via temperature abuse from refrigerant leak for example [30]) without causing an order backlog, it could be considered flexible. Finally, the percentage of shipments that are wasted is included as a key performance indicator (KPI). These are defined in the model as percentage of shipments wasted, $P_{w}$, miles driven per successful shipment, $M_{s}$, the hours spend waiting per successful shipment, $H_{s}$, and the surplus shelf life of successfully delivered shipments, $L_{av}$. $L_{av}$ is used as calculated from the remaining spoilage range, and it is representative of the additional time before the retailer rejects the shipment.

$P_{w}$ is defined by Equation (Equation 1)
(1)$$P_{w}=\frac{S_{w}}{S_{t}}$$
where $S_{w}$ is the number of shipments wasted over the course of the simulation, and $S_{t}$ is the number of total shipments that left the Supplier. A shipment is considered wasted if the Carrier could not be re-routed to a Customer within spoilage range, or if it was delivered to the Customer spoiled. $S_{w}$ is collected by adding 1 to a variable every time an order is cancelled (only available when sensorisation is enabled in the model) or when a spoiled shipment arrives at the Customer. $S_{t}$ is collected by adding 1 to a variable every time a Carrier leaves the Supplier.

$M_{s}$ aims to capture the impact on the resources expended in delivering a successful shipment. If Carriers can be returned to the Supplier for disposal of a spoiled or spoiling shipment before they reach the Customer, then resources (time, fuel, driver wages) could be saved. The metric is defined by Equation (Equation 2)
(2)$$M_{s}=\frac{M_{t}}{S_{s}}$$
where $M_{t}$ is the sum of all the miles driven by all the Carriers, and $S_{s}$ is the number of unspoiled shipments that successfully arrive at a retailer. $S_{s}$ is collected by adding 1 to a variable every time an unspoiled cargo arrives at a Customer. $M_{t}$ is collected using a function. This function adds the distance between a Carrier a point stored in two variables (one for longitude one for latitude) via road to a third variable (keeping track of the total miles driven by all Carriers)—this is achieved using the inbuilt GIS method “getDistanceByRoute()”. It then updates the longitude and latitude variables to the current position of a Carrier. This function is called every time the Carrier destination changes. The function also updates the spoilage range every time it is called. This is done by simply subtracting the value returned by “getDistanceByRoute()” from the current value for spoilage range. Spoilage range is set to a maximum value every time a Carrier loads a new shipment and is a distance calculated using average Carrier speed and expected remaining shelf life at the Supplier.

$H_{s}$ was characterised in the model as the number of hours Customers spent outside of their normal operation state. This metric is defined by Equation (Equation 3)
(3)$$H_{s}=\frac{H_{p}+H_{w}}{S_{s}}$$
where $H_{p}$ is the total number of hours spent placing an order (while there is no available Carriers) by all of the Customers and $H_{w}$ is the number of hours spent in the waiting for a delivery by all of the Customers. $H_{p}$ and $H_{w}$ were both collected by recording the model time whenever a Customer changed states (normal operation to placing an order to awaiting delivery and back again). As an exit action to each state, the difference between the current model time and the model time last recorded was added to one of two global variables. There is a global variable for time placing an order and time awaiting delivery.

The final metric was the mean remaining shelf life of successfully delivered shipments, $L_{av}$. This was measured in the model by summing the remaining spoilage range to get the total remaining spoilage range, $R_{t}$, of all successful shipments. This was then converted to a number of days using the mean Carrier speed, $s_{m}$. The metric is defined by Equation (Equation 4)
(4)$$L_{av}=\frac{R_{t}}{S_{s}}\cdot \frac{1}{1600\times 24\times s_{m}}$$

$R_{t}$ was found by adding the value for spoilage range of each Carrier to a global variable. This was done whenever a Carrier arrived at a Customer with an unspoiled cargo.

## 5. Experimental Procedure

Three scenarios were selected to investigate how different supply chain parameters affect the relative benefit of increased sensorisation. Here, “scenario” specifies a specific combination of parameter values, customer quantity and location, and supplier location. Due to the long simulation time before convergence, requiring manual restart of the AnyLogic simulation and a finite project scope, three scenarios were selected. The first aims to replicate a general supply chain in the UK. To achieve this, customers were placed in areas with high population density based on data obtained by the Office for National Statistics from 2020–2021 [33].

The other two scenarios were selected to test a hypothesis about the effect of customer’s positions relative to each other on the potential for order swapping. Assuming perfectly direct roads, customers arranged in a circle surrounding the supplier will provide no opportunity for swapping. For these situations, sensorisation of the supply chain will not decrease the $P_{w}$. Conversely, customers arranged in a straight line with the supplier placed at one end should yield the maximum potential for swapping. All scenarios used ten customers as parameter variation revealed this causes a tighter grouping of results in a shorter time.

All scenarios shared the same value for “orderFreqLambda” and “modeTruckSpeed”. “modeTruckSpeed” was selected such that the mean value for carrier speed was 60 MPH [35], and “orderFreqLambda” was set to return a mean inter-order time of 4 days. Finally, an initial value for the maximum delivery time was needed so the initial “spoilageRange” of each carrier can be calculated. It was assumed that customers require a remaining shelf life of 7 days when a shipment arrives, as perishable goods cannot be delivered on the brink of spoilage. Meat is susceptible to spoilage, and so its maximum allowable shelf life (13 days) was selected for calculating remaining shelf life [37].

Each scenario was also calibrated using Monte Carlo analysis in AnyLogic Cloud, ensuring the non-digitalised version had a mean $P_{w}$ value of 4±0.2%. The digitalised versions were then run using the same parameters so that accurate values for expected change to performance metrics would be found. For calibration, the number of carriers and spoilage rate (adjusted to a resolution of 0.025) were adjusted simultaneously. This was conducted until increasing carrier number provided no benefit to $H_{s}$.

Data collection began by running the first scenario with digitalisation. Each simulation was run until the values for performance metrics converged. When convergence checking was enabled the performance metrics were sampled every five model days. If all performance metrics were within 0.1% of their previous values for 10 samples (50 days) consecutively, the simulation was stopped.

The simulations were repeated until the mean values for the $P_{w}$ converged. Here, convergence was considered 5 consecutive variations of less than 0.005 in the percentage improvement, $I_{\%}$, to $P_{w}$. $I_{\%}$ was calculated using Equation (Equation 5) where $E_{s}$ is the number of times the “spoiledEarly” state was entered. This process was repeated for the two remaining scenarios.
(5)$$I_{\%}=\left(\frac{E_{s}}{S_{t}}-P_{w}\right)$$

Once digitalised model data had been collected, the number of runs and maximum run time were noted. These values were used in the Monte Carlo analysis employed to obtain non-digitalised data.

## 6. Results

The final comparison between the two sets of collected data is presented in Figure 8. The mean values for performance metrics are presented in Table 5. These results are 12 box and whisker plots that show the distribution of results obtained for each performance metric for each scenario simulated. They show how the variance in performance metrics differs between digitalised and non-digitalised versions of each scenario. These provide additional context to the mean values presented in Table 5—helping to show if a small change in mean value is significant. There appears to be a decrease in each performance metric for scenarios 1 and 2. For scenario 3, there is also a decrease in each performance metric, except for $P_{w}$. This indicates that digitalisation might be more useful when Customers are not arranged in a ring around the supplier, as there is more capacity to save shipments via dynamic re-routing. However, even if no re-routing occurs, there is still a resource-saving benefit from digitalisation.

## 7. Discussion

The effect of modelling assumptions, as well as other sources of inaccuracy, on the results presented above should be discussed. This discussion can then be taken into account when using the results to draw conclusions about how suitable this approach is for the CBA of sensorisation of CSCs.

### 7.1. Implications of Results

How can the presented results be used to draw conclusions about the impact on stochastic costs and benefits as a result of increased sensorisation in the cold food supply chain? The primary stochastic cost that this modelling approach helps to quantify is the cost of cargo lost due to unexpected spoilage.

The change to $P_{w}$, as well as the other performance metrics, is small. The difference in mean values between the digitalised and standard supply chain is 0.33% for scenario 1. This is due to the small percentage of total shipments affected by unexpected spoilage—approximately 4%. An even smaller percentage of those are available for re-routing. Implementation may still be profitable if the product value is sufficiently high.

Figure 9 shows how the results of this modelling approach might be used to draw conclusions about the suitability of digitalisation for the case study chosen. Note that Figure 9 and the following discussion is using values obtained from simulations of scenario 1. Figure 9 shows how for different setup costs (the cost associated with implementing the sensor network) and values of cargo inside each carrier a CSC manager could determine their payback time. The equation used to create the curves shown in Figure 9 is Equation (Equation 6).
(6)$$T=\frac{C_{s}}{\left(V_{c}\cdot N_{s}\cdot I_{w}\right)+S_{d}+S_{w}-C_{h}}$$
where *T* is the payback time in years, $C_{S}$ is the setup cost in GBP, $V_{c}$ is the value of cargo transported by each carrier in GBP, $N_{s}$ is the mean number of shipments leaving the Supplier per year, $I_{w}$ is the mean improvement to the percentage of cargo wasted per year, $S_{d}$ and $S_{w}$ are the savings on diesel and driver wages per year in GBP/year, respectively, and $C_{h}$ is the cost of the IoT hub required for sensorisation per year in GBP/year. The values for fixed parameters used in Equation (Equation 6) are shown in Table 6.

Figure 9 is an example of how this modelling approach can be used to quantify stochastic properties of digitalised CSCs before sensorisation has been implemented. It uses the improvement to $P_{w}$ and $M_{s}$ (identified as variable/stochastic costs and benefits in Figure 1) to help determine the payback time of a sensor based approach to reducing distribution waste using dynamic re-routing.

This modelling approach can also provide insights into the intangible costs and benefits of sensorisation for dynamic re-routing. For the performed case study, these costs/benefits are the impact on customer satisfaction and product quality. The collection of the performance metrics $H_{s}$ and $L_{av}$ enable the user to investigate the impact of re-routing on customer wait times and the remaining shelf life of delivered produce, respectively. For the considered case study, there is small but clear decrease in wait times. This can be attributed to the ability to order a fresh shipment before the original shipment arrives in a spoiled state. There is no clear impact on product quality, although some shipments may have a decreased quality if they were going to spoil but were redirected to a nearer retailer. These shipments will make up a small percentage of total delivered shipments, so seem to have a negligible impact on $l_{av}$.

The simulation considered scenarios with four carriers. If the cost of outfitting these Carriers with the required connectivity hardware combined with other setup costs was approximately GBP 25,000 and the cargo was a large shipment of beef steak, valued at around GBP 1000, then the payback time would equal 4.2 years. A payback time of 5 years can be considered acceptable for supply chain investments [43], so sensorisation for dynamic re-routing may be appropriate for implementation into the supply chain discussed here. These calculations highlight the advantages of simulation for CBA. Even though the detected change in performance metrics is small, implementation may still be a worthy investment. Since small changes in these stochastic costs and benefits remain important, efforts should be made to quantify them. However, it is assumed that changes in customer wait times in the order of minutes are unimportant, and hence are not included in the cost–benefit calculations. Supply chains where wait times might play a more significant role in cost–benefit calculations are those where the product is consumed at the point of delivery. Examples might include the distribution of donor organs, foodstuffs where the freshness comprises most of the value (fresh caught fish in high-end restaurants), or perhaps components used in just-in-time production lines.

If a maximum allowable payback time is decided (5 years for example) then this approach can also be used to calculate the minimum required value of transported product—such that sensorisation remains a worthwhile investment. Once again, considering a setup cost, $C_{s}$ of GBP 25,000 and using the values for parameters listed in Table 6, Equation (Equation 7) can be used to find the minimum required cargo value, $V_{min}$. This returns a minimum cargo value of GBP 682 for the given scenario.
(7)$$V_{min}=\frac{\frac{C_{s}}{T}-S_{d}-S_{w}+C_{h}}{N_{s}\cdot I_{w}}$$

A product with a much higher value is more likely to benefit from sensorisation. In fact, using Figure 9 and the examination of different case study scenarios, several conclusions can can be drawn about what type of CSCs are more likely to benefit from sensorisation. These conclusions are other examples of how this modelling approach can be used to gain insight into other indefinite costs and benefit. They are:CSCs with a low initial cost for sensorisation can expect more benefit. This could mean CSCs whose product quality can be accurately monitored using low cost sensor networks. Implementations that use cheaper sensors, such as temperature probes, and can handle more infrequent sampling times, therefore needing more lightweight connectivity hardware, will have lower setup costs [44]. Further, infrequent sampling times will reduce the subscription costs for information transfer network costs if fewer messages are required per day [39].By comparing data between scenarios 1, 2, and 3, it appears that a supply chain sees the greatest reduction in spoilage when Customers are arranged in a line, and no reduction when arranged in a circle. This is likely due to a line arrangement offering a larger number of re-routing options. Therefore, supply chains with a similar arrangement and the supplier at one end of the line could expect a larger benefit.

A currently relevant supply chain that could be well suited to this approach would be the distribution of vaccines during a pandemic. They are developed rapidly in response to an event. As such, the supply line may not be well established. This provides the opportunity for strategic network arrangement to minimise the chance of wastage. They are required internationally, so the benefit to $H_{s}$ and $M_{s}$ would be more significant—important as they may be urgently required. Further, they may require careful storage conditions and have short shelf lives [45]. Therefore, they could provide more opportunities for re-routing as spoilage may be more frequent. Finally, the benefit of successful re-routing is far beyond the cost of production, as it can prevent death and illness.

Moving forward, researchers may wish to investigate several areas:Inclusion of spoilage models for perishable cargo in the distribution model. This would supplement a stochastic trigger, causing quality abuse. This might mean a trigger causing temperature abuse that affect the quality of the cargo using the spoilage model. This may influence where spoilage occurs during distribution, leading to more accurate results.Including different or multiple optimisation criteria when re-routing. In this work, only driven distance was considered, but in reality, vehicle wear and driver availability would be important factors.This approach could be used to investigate the relative benefits of different sensor architectures, specifically their sample rates. Different scenarios with different inter-spoilage times could be run to investigate whether a more expensive sensor network capable of more frequent sampling is worthwhile.This work only considers re-routing as a possible use for sensorisation. Other options that use the same sensors may exist, for example, dynamically adjusting the refrigeration.The impact on customer wait times and product quality could be used to make more qualitative recommendations. For instance, focus group studies could be employed to evaluate the acceptable levels of delay and product deterioration and to identify areas for improvement. By gathering detailed feedback from key stakeholders, these studies could inform the development of more effective supply chain strategies that prioritize both efficiency and quality.

### 7.2. Inaccuracy in Results

The assumptions listed in Table 4 may impact how accurate the results are for the chosen case study. When combined with the case study assumptions, these also limit the scope of the conclusions that can be drawn about CBA. The inaccuracies caused by assumptions are discussed here in Table 7.

## 8. Conclusions

This study used a multi-method modelling approach to assess the potential costs and benefits of increased sensorisation in a CSC. The study combined agent-based and discrete event modelling to simulate digitalised and non-digitalised versions of different supply chain arrangements. The study presented a simplified case study of beef distribution in the UK and analysed the effects of dynamic re-routing on supply chain performance metrics. Using the simulation results, the study provided recommendations for whether sensorisation for dynamic re-routing should be adopted in the case study supply chain based on payback time and effects on intangible supply chain costs and benefits.

This work has shown that advanced modelling and simulation can be a useful approach for the cost–benefit assessment of digitalisation in cold distribution chains. It has the ability to predict stochastic costs and benefits, such as the impact on cargo wasted, resources expanded in delivery, customer wait times, and product quality. These can be used to make predictions about payback time for digitalisation with greater accuracy, as current methods for cost–benefit analysis of digitalisation neglect these criteria. This is significant as we move towards Industry 4.0 and an increasing portion of industry begins to consider increased sensorisation as a efficiency improving measure. It is particularly relevant for the CSCs moving perishable products, such as food and medication, as an expanding population, environmental concerns, and sudden increases in demand place pressure on them.

## Figures and Tables

**Figure 1 sensors-23-04147-f001:**
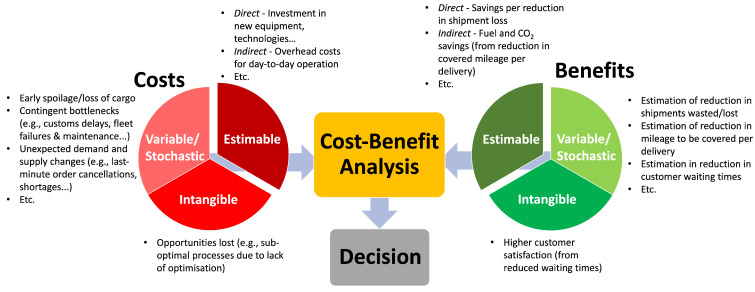
A figure showing the difficulties when considering the cost–benefit of digitalisation in CSCs.

**Figure 2 sensors-23-04147-f002:**
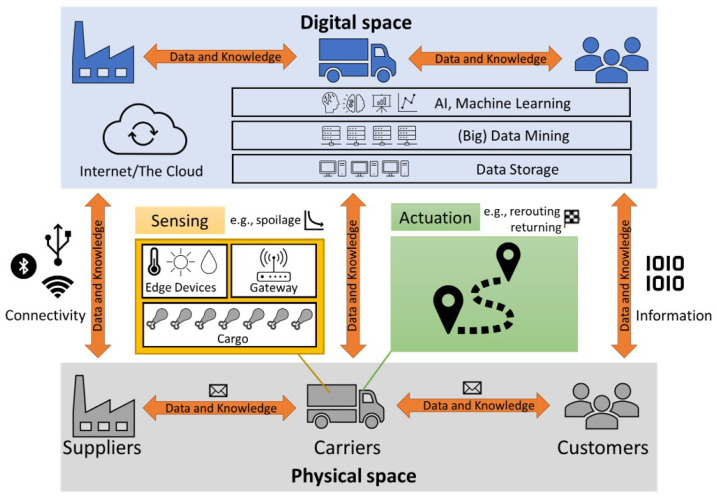
The conceptual model of a general digitalised distribution cold chain.

**Figure 3 sensors-23-04147-f003:**
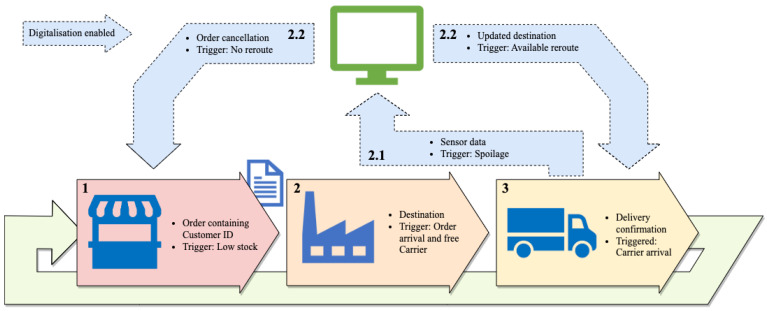
A Communication between agents during simulation. The numbers represent the order of communications in the simulation. 2.1 and 2.2 do not always occur, only when spoilage is detected. There are two variations of step 2.2 as they cannot simultaneously occur.

**Figure 4 sensors-23-04147-f004:**
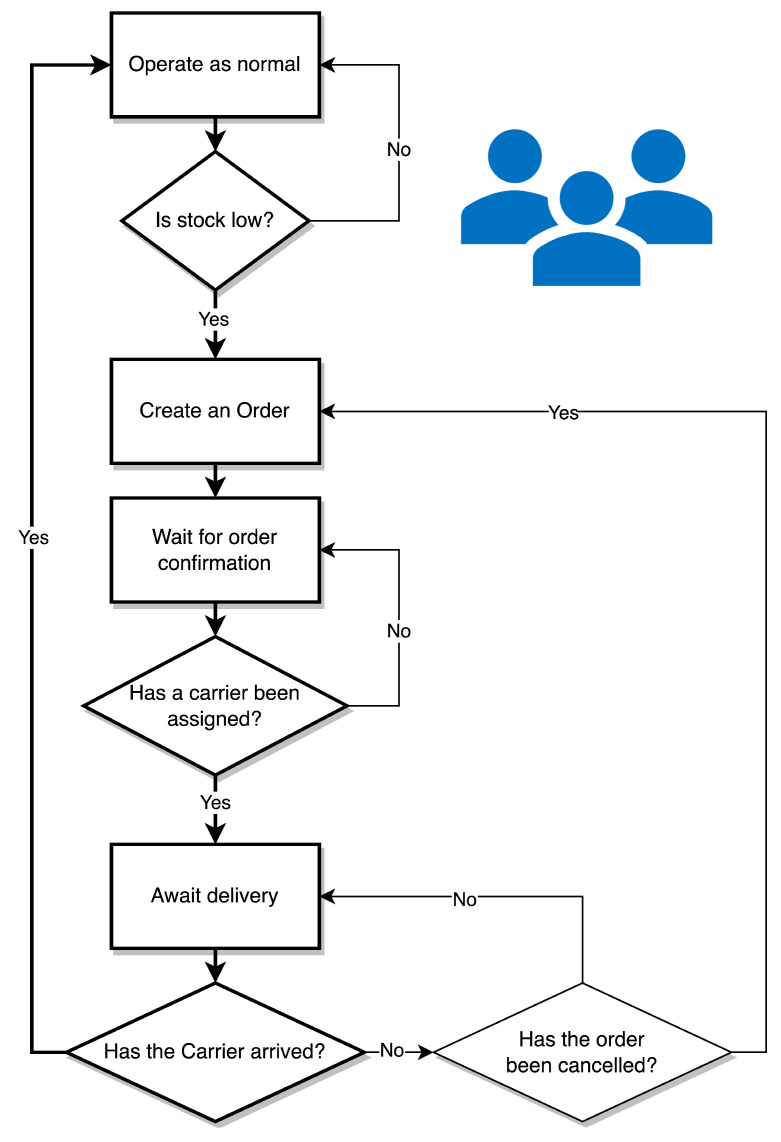
Customer agent statechart.

**Figure 5 sensors-23-04147-f005:**
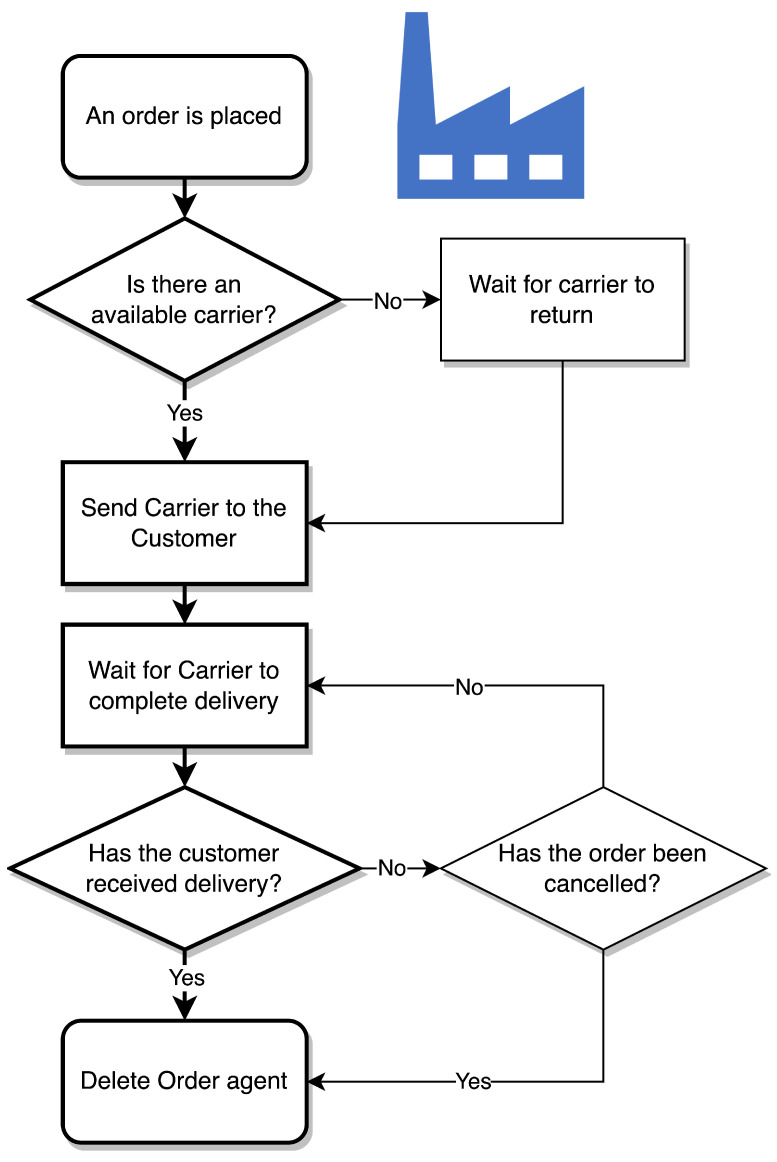
Supplier agent statechart.

**Figure 6 sensors-23-04147-f006:**
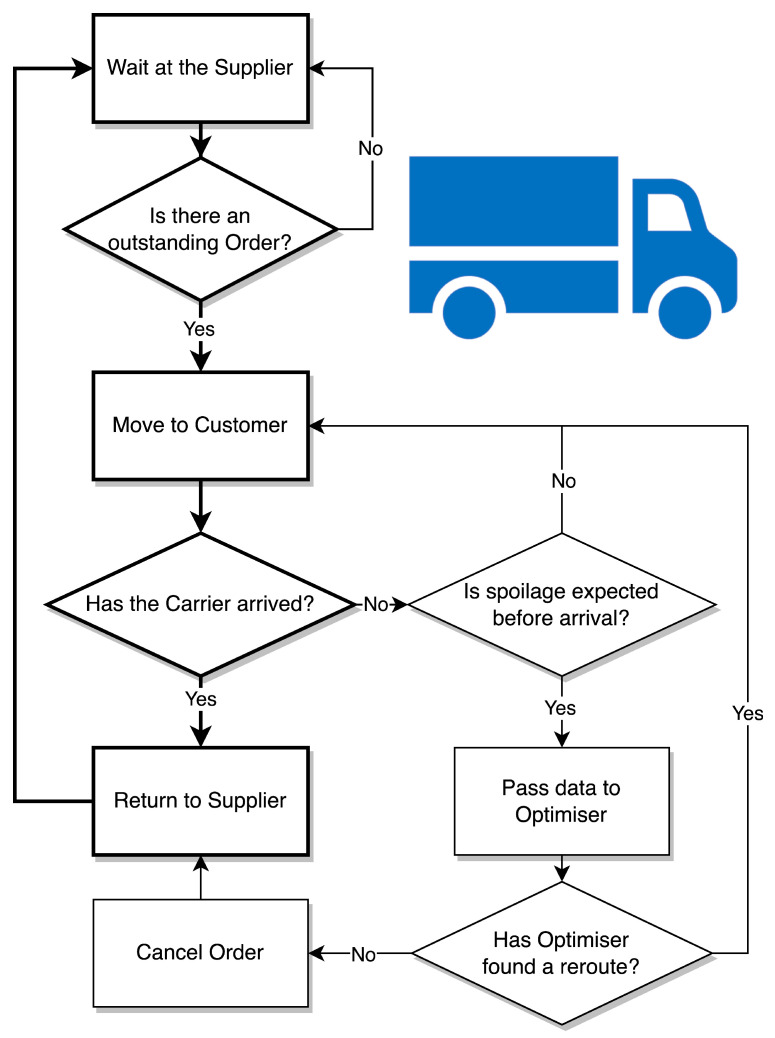
A flowchart describing the behaviour of the Carrier agents.

**Figure 7 sensors-23-04147-f007:**
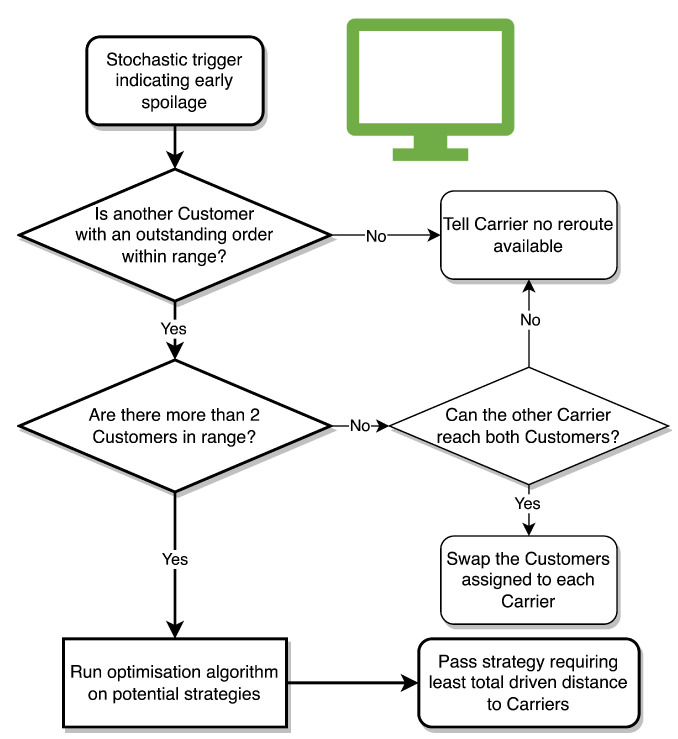
The logic followed by the Optimiser.

**Figure 8 sensors-23-04147-f008:**
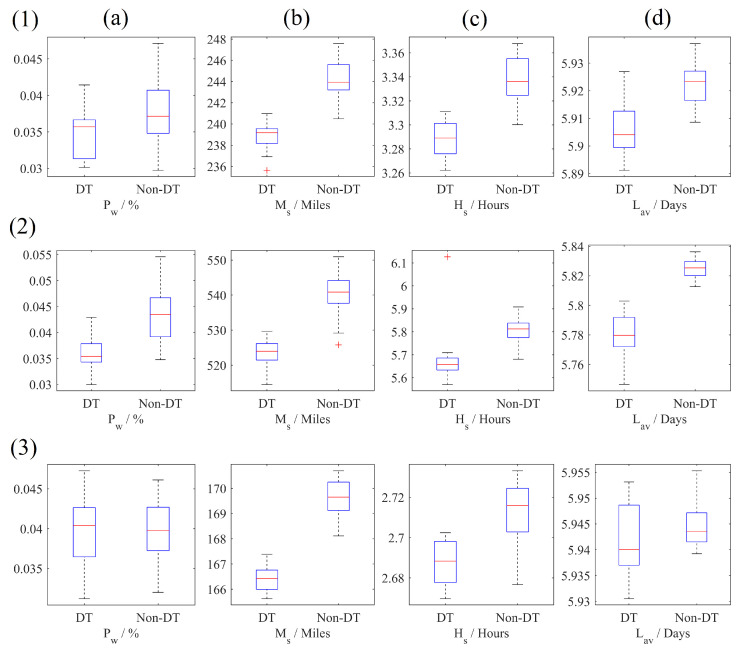
Box plots representing the distribution of performance metrics $P_{w}$, $M_{s}$, $H_{s}$, and $L_{av}$ (**a**, **b**, **c**, and **d**, respectively) for scenarios 1, 2, and 3. The largest disparity between digitalised and non-digitalised runs is seen for scenario 2, indicating that it is more likely to benefit from increased sensorisation. Here, DT represents digitalised. Note that axis labels follow the format performance metric/unit. For example, $H_{s}$/Hours is the time spent waiting for a delivery by Customers measured in hours.

**Figure 9 sensors-23-04147-f009:**
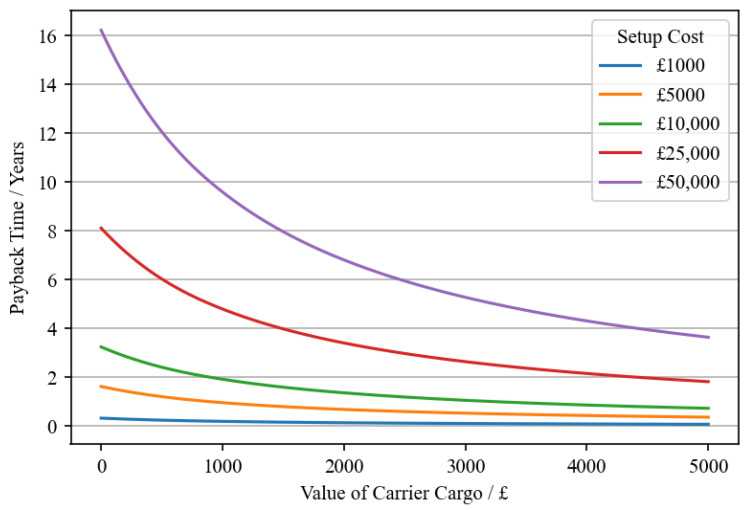
A figure showing how payback time varies for different values of carrier cargo for different setup costs, which is relevant for scenario 1 of the case study chosen for this work.

**Table 1 sensors-23-04147-t001:** The benefits of AnyLogic as the simulation package used for this work.

#	Feature	Utility
1	Multi-method modelling	Allows models to combine DES and ABM, both of which are required for the described modelling approach [14,16]. Single method approaches will require cumbersome workarounds [31].
2	Integrated geographic information system (GIS) functionality	Enables easy and accurate calculation of travel distances between agents, providing an easy metric to base optimiser decisions on. Also enables effortless creation of numerous pathways between agents. This makes the creation of unique options for vehicle re-routing simple.
3	Established simulation	Error detection is made much easier than in a model created in a general purpose programming language. In addition, a natural framework for model construction is provided, reducing the time and effort required to adopt this approach [32].
4	Easy implementation of stochastics/statistical distributions	Generation of a vast number of unique scenarios is made simple. Representation of stochastic events such as spoilage and order placement is easy.

**Table 2 sensors-23-04147-t002:** The benefits of Microsoft Excel as the software package for creating the Optimiser.

#	Excel Feature	Benefits When Applied to This Work
1	Ubiquity of Excel in industry	The optimisation method described in this work can be easily adopted by most readers.
2	Application of heuristics through Solver	The Solver add-in for Excel makes the application of a variety of heuristics simple. This means the structure of the optimisation problem is not such a concern.
3	Compatibility through Visual Basic for Applications (VBA)	The Solver add-in, as well as other features, are compatible with VBA. This will make the execution of the Optimiser during the simulation more streamlined and ease data collection.

**Table 3 sensors-23-04147-t003:** Conversion of the general modelling terminology/description into an AnyLogic specific format.

Number	Generalised Terminology/Feature	Case Study in AnyLogic Terminology/Feature	Quantity
1	Customers	UK beef vendors (supermarkets, butchers, restaurants) placed according to population density.	10
2	Suppliers	A beef packaging facility located on the Welsh border.	1
3	Carriers	Fleet of delivery trucks.	4
4	Orders	Order of beef. Order size is not considered.	-
5	Optimiser	Evolutionary algorithm applied using Solver add-in.	-
6	Carrier data	GPS co-ordinates and shelf life of shipment.	-
7	Stochastic spoilage trigger	A rate for spoilage that calibrated the non-digitalised supply chain to have a spoilage rate of 4% [34].	-
8	Stochastic order trigger	An exponential distribution with a mean of 1 week.	-

**Table 4 sensors-23-04147-t004:** Assumptions made in order to model the supply chain.

#	Assumption
1	All deliveries are carried out using the UK’s road network.
2	Truck fleet is homogeneous, providing equal refrigeration, having an equal mean speed of 60 MPH [35].
3	Trucks do not wear, and driver constraints are not considered.
4	Disposal of spoiled shipments occurs at the Supplier and is instant.
5	Suppliers have an unlimited stock.
6	Sampling frequency of sensor network is high, so shipments can spoil anywhere along their journey.
7	Shipments spoil entirely and there is no option for partial delivery.
8	The causes of spoilage are unimportant and not distinguished.
9	Order size is not considered and there is only one type of product.
10	Orders are always of the same, maximum quality/shelf life at the Supplier.

**Table 5 sensors-23-04147-t005:** The mean values of the data presented in Figure 8 where DT represents digitalised.

Scenario	$P_{w}$ [%]		$M_{s}$ [Miles]		$H_{s}$ [Hours]		$L_{\mathrm{av}}$ [Days]	
	DT	Non-DT	DT	Non-DT	DT	Non-DT	DT	Non-DT
1	3.48	3.81	238.84	244.25	3.29	3.34	5.91	5.92
2	3.57	4.19	523.33	540.33	5.67	5.81	5.78	5.78
3	3.97	3.94	166.39	169.59	2.69	2.71	5.94	5.94

**Table 6 sensors-23-04147-t006:** Fixed values used in Equation (Equation 6) relating to scenario 1 for the chosen case study of beef distribution in the UK. Note that the cost of IoT hub, C${}_{h}$ is zero. In the worst case scenario, where all Carriers are all out for delivery and transmitting data, the required number of messages per day still falls within the free tier of the Azure IoT hub.

No.	Parameter	Value	Unit	Used to Find	Source
1	Number of Carriers	4	N/A	3	Table 3
2	Sampling Frequency	1	minutes${}^{-1}$	3	[38]
3	Maximum possible Messages per Day	5760	days${}^{-1}$	Cost of IoT hub	Calculation
4	Cost of IoT hub, $C_{h}$	0	GBP/year	Payback time	[39]
5	Mean Number of Shipments per Year, $N_{s}$	852	years${}^{-1}$	9, 14, and Payback time	Simulations
6	Improvement in $P_{w}$, $I_{w}$	0.33	%	Payback time	Table 5
7	Carrier Miles per Gallon	18.78	MPG	10	[40]
8	Mean Miles Saved per Successful Delivery	5.41	miles	9, 13	Table 5
9	Mean Miles Saved per Year	4607.16	miles/year	11	Calculation
10	Price of Diesel per Mile	0.4072	GBP/mile	11	[41]
11	Savings on Diesel, $S_{d}$	1876.15	GBP/year	Payback time	Calculation
12	Mean Truck Speed	60	MPH	13	Table 4
13	Driving time saved per Successful Delivery	0.09	hours	14	Calculation
14	Time Saved per Year	76.79	hours/year	16	Calculation
15	Truck Driver Hourly Wage	15.73	GBP/hour	16	[42]
16	Savings on Driver Wages, $S_{w}$	1207.84	GBP/year	Payback time	Calculation

**Table 7 sensors-23-04147-t007:** Discussion of each assumption and how it might cause inaccuracy in the results.

#	Sources of Error	Effect
1	The GIS functionality in AnyLogic is accurate and any inaccuracies in delivery distance are likely to be negligible.	N/A
2,3	In reality, trucks within a delivery fleet would differ from each other in a number of ways: vehicle wear, driver ability/choices, model, and size. The most significant of these factors is model and size. Different models may have different fuel efficiencies and accelerations. However, due to the average delivery time being far greater than refuelling times and routes often using motorways, the model of truck can be ignored. The size of the truck may limit the orders that it can take, which may reduce the number of available trucks at times. However, since order size was ignored; this will not introduce an inaccuracy in the results. All trucks were assigned a speed of 60MPH. In reality, their average speed would be slower due residential areas along the delivery route, refuelling, etc. Readers seeking to implement this approach might be able to obtain a more accurate value for the average speed of their carriers, reducing this inaccuracy.	$H_{s}$ may be slightly longer in reality and $L_{av}$ may be slightly shorter.
4	This assumption simplifies the scenario by reducing the number of actors and enables trucks containing spoiled cargo to be available as soon as disposal was completed. If a disposal centre was located elsewhere or disposal was not instant, then this may reduce the average number of available trucks, extending time spent waiting for an order and increasing the miles driven by each truck.	$H_{s}$ and $M_{s}$ may be increased.
5	A Supplier running low on stock, in this case beef, may be caused by a shortage of that product. If Suppliers limit the amount of orders they promise to deliver, then none of the PMs should be affected. $H_{s}$ would only be increased if Suppliers confirm orders that they do not have the stock for.	N/A
6	The sampling frequency of the sensor network required to support the data collection used for dynamic re-routing may range from seconds to minutes or more depending on power requirements, data storage, connectivity, etc. In reality, it would most likely be less than several minutes [38]. The implication of this is that spoilage may occur between samples and cause a truck to take a slightly less optimal route. This might have a small negative affect on all PMs.	A small to negligible negative effect of all PMs.
7	Order size is not considered in this model; however, partial delivery/spoilage could be considered if it acted as a modifier to customer’s time between orders, and was included in strategy optimisation. If this level of abstraction was chosen, then the percentage of shipments wasted should be reduced as well as time spent waiting for an order. It maybe increases the miles driven to deliver the same amount of product, and the average surplus shelf life of delivered product would decrease.	Increased values of $H_{s}$. Decreased values of $P_{w}$, $M_{s}$, and $L_{av}$.
8	The model was calibrated such that the values for spoilage during distribution match those found in literature. Since order size is also not considered, then the cause of spoilage should not provide different strategy options even if it was considered, as there would be no option for partial delivery if a single piece of packaging was faulty, for example. The cause of spoilage may affect the remaining shelf life of products if it was a temperature abuse affecting all the Carrier’s cargo, for example. However, if shipments spoil for any reason, they would not be included in the PM, so considering spoilage causes still would not affect PM accuracy.	N/A
9	If order size were considered, then this might substantially reduce the options for re-routing, unless partial delivery was also allowed. In either case, this would probably increase the amount of miles driven to deliver the same amount of product. It might increase the hours spent waiting and the amount of cargo wasted if partial delivery is not allowed. The surplus shelf life might increase if there is not partial delivery, as more often deliveries will have to come straight from the Supplier and not via a re-route.	If partial delivery is not also included, then all PMs may increase.
10	If cargo had varying quality at the supplier, then the amount of wasted cargo can only increase as currently it is always of maximum quality. This should also increase the miles driven and hours spent waiting per delivery. It should also reduce the surplus shelf life, as product will start with a smaller shelf life.	Increased values of $P_{w}$, $M_{s}$, and $H_{s}$. Decreased values of $L_{av}$.

## Data Availability

Publicly available datasets were analyzed in this study. This data can be found here: https://github.com/OliverSchiffmann/CBA-for-CSC (accessed on 27 February 2023).

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
