# Peer review of "A Cost–Benefit Analysis Simulation for the Digitalisation of Cold Supply Chains"

_sensors, 2023, doi:10.3390/s23084147_

Round 1
Reviewer 1 Report
The problem definition and contribution itself are not stated explicitly. It is challenging to pinpoint the precise problem the authors want to solve. In addition, I believe that the authors are attempting to analyze the cost-benefit of implementing digitalization in cold supply chains. It is likely that the authors want to attract readers who are small and medium-sized enterprises (SMEs). However, the case study shows that the impact of digitalization on SMEs is negligible. It's unclear if the authors are trying to contribute a new approach or prove something. Furthermore, the paper's calculations, data, and experiments are poorly explained, making them difficult to understand. In addition, the insufficient case may lead to a lack of explanation of where the advantage and limitations of the proposed approach. My detailed comments are as follows:
Major issues
-
The problem definition and contribution itself are not stated explicitly. It is challenging to pinpoint the precise problem the authors want to solve. In addition to Industry 4.0, this topic could also be categorized as Logistics 4.0. It addresses the many logistical aspects of optimization between human and digital ecosystems.
-
It's unclear if the authors are trying to contribute a new approach or prove something.
-
“Carrier agent only handles one Order agent at a time” and “allowed to reroute and change Customer agent.” (lines 287-288). What makes this possible?
-
Not taking into account: customers' orders and quantities, stated time of arrival, and customers' wait time. As a result, the cost-benefit analysis might not be entirely convincing.
-
Table 5 shows very minor improvements in which the case received no real advantages. Is the paper implying that the impact of digitalization is negligible in SMEs?
-
What are FSC and RSL?
-
Does this suggest that route planning before delivery is more crucial than during delivery?
-
“A shipment is considered wasted if the truck could not be re-routed to a Customer within spoilage range, or if it was delivered to the Customer spoiled” (lines 377-379).
-
In the current situation, what are the critical issues related to delivery across the UK and spoilage? For instance, it might spoil inside the refrigerated truck if it is transported too far or for too long. However, How long does delivery typically take in the UK?
-
Insufficient case study.
-
“... the study provided recommendations for whether sensorisation for dynamic re-routing should be adopted … based on payback time and effects on intangible supply chain costs and benefits” (Lines 582-584). The findings, however, point to a scenario with insignificant improvements and a longer payback period.
-
“This study is well suited to the distribution of vaccines during a pandemic.” (lines 540–541). Why not incorporate this scenario as well?
-
This paper should demonstrate a situation in which the proposed approach has a significant impact (contribution). It is also strongly advised to include a study case where the proposed approach has a negligible impact (limitation).
-
The calculation doesn't seem to be very credible.
-
I believe that payback time would take into account the "fixed and variable cost" and "contribution margin per unit." However, in Table 6, the "Ch," or the cost of the IoT hub, is set to £0/year (zero cost). Whereas, the use of the phrase "subscription costs" in line 533 seems to contradict it.
-
“Wales was chosen as the largest meat supplier to major UK cities such as London, Liverpool, and Manchester …” (317 - 319). However, there is no information available on the average delivery time in the UK. As a result, the simulation is not clearly explained.
-
There are discrepancies between “Pw = 4%” (Lines 441 and 445) and “Pw = 4 ± 0.2%” (446-447). It needs to be clarified (or explained) more because readers might be confused.
-
The phrase "payback time with greater accuracy" exists, but it lacks validation and a baseline against which to compare.
Minor issues
-
Figure 8 is most likely displaying inconsistent and misleading labels. E.g., Hs/hours will be read as how many "hours per hour," while Lav/days will be read as surplus shelf life per day (?)
-
Subsection 4.1.1. is too short, consider merging it into another section.
-
Line 532 contains question marks ("??").
Reviewer 2 Report
Dear author,
Article is well written with interesting topic and good simulation research. Here are some little remarks:
· Explanation in Figure 2 and 3 put in the text, not in the name of table.
· Line 160: space before comma is missing.
· Line 405: is there 0 necessary?
· Line 510: comma is put in wrong place.
· Line 532: Question marks ??. Remove it.
Best regards, reviewer
Reviewer 3 Report
This manuscript focuses on the distribution of refrigerated beef in the UK, where digitalisation was implemented to re-route cargo carriers. By comparing simulations of both digitalised and non-digitalised supply chains, the study found that digitalisation can reduce beef waste and decrease the number of miles driven per successful delivery , leading to potential cost savings. The application of digital technology to industry has innovative and practical significance. By providing a better understanding of the potential costs and benefits of digitalisation, simulation can help organizations make more strategic and effective decisions. The technical introduction and reasoning in the manuscript are rigorous .I think the manuscript can be published on Sensors.
Reviewer 4 Report
The article is current because the research in the article are of great importance for the study of investigates the potential benefits and costs of digitalising cold distribution chains in the food supply chain using simulation. For the management of organizations, such a study is important from the point of view of strategic decisions as well as for a better understanding of the benefits of digitalization to existing systems. My comments on the article are the following:
1. Figures 1 and 3 – the texts in the figures are not easy to read, they are too small. They have to be enlarged and the figures have to be put into a readable form.
2. In the article, the authors mention line 69 effective implementation strategies. Do the authors have in mind specific implementation strategies in the supply chain or are they talking about implementation strategies in general.
3. In line 124, the authors write that "The optimal cold storage location and shipping quantities were determined." Did the authors mean the cold storage location and shipping quantities specifically in the trailer of the truck being transported? Doesn't the optimal cold storage location depend on the type of goods being transported?
4. In Chapter 2.3, specifically in line 150, the authors write about the environmental factors of the cargo. It would be good if the environmental factors of the cargo were explicitly listed in this Chapter.
5. Figure 3 can be found in chapter 3.1. Conceptual Model of a General Digitalized Cold Distribution Chain. However, the textual description of Figure 3 is in Chapter 3.2 Model operation. For a better understanding of the meaning of the text and a better chronology, it would be better for the authors to move Figure 3 to Chapter 3.2, line 218.
6. In Chapter 3.3.1 The Customer Agents, the text refers to figure no. 4. However, this figure is found in Chapter 3.3 of The Carrier Agents. For a better understanding of the meaning of the text and a better chronology, it would be better for the authors to move Figure 4 to Chapter 3.3.1 on line 260.
7. In Chapter 3.3.2 The Supplier Agents, the text refers to figure no. 5. However, this figure can be found in Chapter 4.1. Case Study - UK Beef CSC. For a better understanding of the meaning of the text and a better chronology, it would be better for the authors to move Figure 5 to Chapter 3.3.2 on line 272.
8. In Chapter 3.3.3 The Carrier Agents, the text refers to figure no. 6. However, this image is found in Chapter 4.2. Optimizer Implementation. For a better understanding of the meaning of the text and a better chronology, it would be better for the authors to move Figure 6 to Chapter 3.3.3, line 284.
9. In Chapter 4, the authors describe the benefits of choices, which are in table 1 and 2. However, these tables can be found in chapter 4.3. Performance metrics. For a better understanding of the meaning of the text and a better chronology, it would be better for the authors to move tables 1 and 2 to the end of Chapter 4. in line 307.
10. The authors write in line 309 that "Meat products are safety-critical foods that have significant economic impact due to their high retail price, a large ecological footprint (e.g., water and land usage), fuel consumption in transportation and CO2". What did the authors mean by the phrase safety-critical foods? Could the authors clarify the term?
11. In Chapter 4.1 Case Study - UK Beef CSC, the authors refer to Tables 3 and 4. However, these tables can be found in Chapter 4.3. Performance metrics. For a better understanding of the meaning of the text and a better chronology, it would be better for the authors to move tables 3 and 4 to the end of Chapter 4.1 on line 322.
12. In line 331 the authors write that "The statechart for the optimiser is shown in 7". It should be added here that it is a Figure 7 and not Section 7.
13. The authors write on line 367 that "If the system can handle unexpected events that cause extended delivery times (spoilage) without causing an order backlog, it could be considered flexible." In the article, the authors should specify which unexpected events may be involved.
14. In line 436, the abbreviation RSL is mentioned. It would be good to explain in the text what this abbreviation stands for.
15. In Chapter 6 results, the authors refer to table 5. However, the table can be found in Chapter 7.1. Implications of results. For a better understanding of the meaning of the text and a better chronology, it would be better for the authors to move table 5 to Chapter 6.
16. A figure 9 showing how payback time varies for different values of carrier cargo for different setup costs (for scenario 1). Could the authors say up to what value of transported goods it is worth to perform sensorisation in CSC for individual scenarios?
17. I recommend including Table 7 at the end of Chapter 7.2 Inaccuracy in Results.
Round 2
Reviewer 4 Report
I have reread the corrected and supplemented article. The authors of the article have given due consideration to my comments and suggestions. The authors of the article have improved the graphic level of the figures. Figures and tables have been arranged chronologically according to their occurrence in the text. An improvement in the quality of the reviewed article can be seen. I must state that the article in its present form is written in an understandable way and I believe that it will be useful for the readers of Senzor journal. I wish the authors success in their scientific work.